# Immune Checkpoint Inhibitor (ICI) Genes and Aging in Clear Cell Renal Cell Carcinoma (ccRCC): Clinical and Genomic Study

**DOI:** 10.3390/cells11223641

**Published:** 2022-11-17

**Authors:** Abdullah Al-Danakh, Mohammed Safi, Mohammed Alradhi, Qiwei Chen, Salem Baldi, Xinqing Zhu, Deyong Yang

**Affiliations:** 1Department of Urology, First Affiliated Hospital of Dalian Medical University, Dalian 116021, China; 2Department of Respiratory Diseases, Shandong Second Provincial General Hospital, Shandong University, Jinan 250023, China; 3Department of Urology, The Affiliated Hospital of Qingdao Binhai University, Qingdao 266000, China; 4Research Center of Molecular Diagnostics and Sequencing, Axbio Biotechnology (Shenzhen) Co., Ltd., Shenzhen 518057, China; 5Department of Surgery, Healinghands Clinic, Dalian 116021, China

**Keywords:** age, immunosenescence, immune checkpoint inhibitors (ICIs), clear cell renal cell carcinoma, TNFSF15, overall survival

## Abstract

**Background:** It is anticipated that there will be a large rise in the number of tumor diagnoses and mortality in those aged 65 and older over the course of upcoming decades. Immune checkpoint inhibitors, often known as ICIs, boost immune system activity by selectively targeting ICI genes. On the other hand, old age may be connected with unfavorable results. **Methods:** The Cancer Genome Atlas (TCGA) provided gene expression data from ccRCC tissue and key clinical variables. ICI gene databases were applied and verified using the GEO database. **Results:** We identified 14 ICI genes as risk gene signatures among 528 ccRCC patients using univariate and multivariable cox hazard models, and the elderly group was linked with poor survival. Then, by utilizing a new nomogram method, the TNFSF15 gene and age predicting values were estimated at one, three, and five years (85%, 81%, and 81%), respectively, and our age-related risk score was significant even after multivariable analysis (HR = 1.518, *p* = 0.009, CI = 1.1102.076). TNFSF15 gene expression was lower in elderly ccRCC patients (*p* = 0.0001). A negative connection between age and the TNFSF15 gene expression was discovered by correlation analysis (*p* = 0.0001). The verification of the gene by utilizing GEO (GSE167093) with 604 patients was obtained as external validation that showed significant differences in the TNFSF15 gene between young and elderly patients (*p* = 0.007). Additionally, the protein–protein interactions of the TNFSF15 gene with other ICI genes and aging-related genes was determined. In addition, the TNFSF15 expression was significantly correlated with pathological stages (*p* = 0.018). Furthermore, it was discovered that the biological processes of senescence, cellular senescence, the immune system, and many immune cell infiltration and immune function types are all closely tied. **Conclusions:** Along with the risk score evaluation, the ICI gene TNFSF15 was identified as a tumor suppressor gene related to inequalities in age survival and is associated with pathological stages and different immunity statuses. The aging responses of ccRCC patients and related gene expression need further investigation in order to identify potential therapeutic targets.

## 1. Introduction

Renal cell carcinoma (RCC) represents the most common malignant solid cancer of the kidney. It is reported that more than 30% of RCC patients are metastatic at the time of the diagnosis, and nearly 30% will progress to metastasis during the course of follow-up [1,2]. Among the three main RCC histological subtypes, clear cell renal cell carcinoma (ccRCC) accounts for about 70% of all RCC cases [1,3]. Surgical excision is the preferred treatment for ccRCC, but the disease’s prognosis is dismal and expected to recur. In the recent few decades, the metastatic renal cell carcinoma (mRCC) treatment has had tremendous progression, starting with high-dose interleukin 2 (IL-2) and interferon-α, followed by targeted therapy such as vascular growth factor receptor inhibitors (VEGF-R), and mammalian target of rapamycin (mTOR) inhibitors [4]. Immunotherapy has finally evolved in the treatment of ccRCC, either alone or in conjunction with other regimens, with promising outcomes. In recent years, four randomized trials investigated immune checkpoint inhibitors (ICIs) either alone or ICIs/VEGF/VEGF-R targeted combination: ipilimumab and nivolumab in the clinical trials of CheckMate 214, or pembrolizumab and axtinib in other clinical trials called KEYNOTE-426, also avelumab and axitinib in JAVELIN Renal 101, and finally atezolizumab in combination with bevacizumab in IMmotion151 [5,6,7,8]. Therefore, it was evident that the integration of ICI-based treatment into the first-line therapy for mRCC had a profound impact on the lives of mRCC patients [4,9].

Immunosenescence is a topic that has lately been brought to the forefront of the attention of the medical community, which has given rise to new lines of investigation as well as debate [10,11,12,13]. Senescence of the cells is an eternal cell cycle arrest that often occurs in cell proliferation in response to different stressors. Senescence can occur under various physiological and pathological conditions, including tissue regeneration, tissue damage, aging, and tumor development [14]. According to a number of studies, the presence of senescent cells in tissues may facilitate the proliferation and invasion of pre-neoplastic cells in the nearby area [15,16]. In general, the effects of cellular senescence on a tumor are extraordinarily complex, playing both positive and negative roles in the genesis of tumors, their recurrence, and the efficacy of treatments [17]. Aging is distinguished by a steady reduction in function, leading to increasing degeneration and tissue malfunction [18]. Aging is revealed as a strong prognostic marker for shorter survival in many cancers and carcinogenic aggressiveness [13,19].

Even though the majority of available studies are cross-sectional and can, therefore, only detect disparities as opposed to real change, they seldom prove with evidence the age-linked differences of the immunological parameters, efficacy, and safety in the elderly which prevents them from immunotherapy administration [20,21,22,23]. The true significance of such differences is still unclear. Therefore, an in-depth investigation of the immunosenescence status utilizing biomarkers that could reduce the knowledge gap of age disparities in ccRCC, particularly in the era of ICIs, is necessary. Integrating public datasets from TCGA and GEO in order to investigate how changes in genomic expression occur with age is becoming an increasingly important field of research.

Our research in this study will establish the molecular status of patients with ccRCC. We will apply genetic profiling of ccRCC cohorts using the TCGA database to identify age-related ICI gene expression. To accomplish this, we will match newly approved ICI genes to the transcriptomic gene data from ccRCC cohorts. In addition, the GEO database will be used to validate the TCGA-generated genes that are associated with aging.

## 2. Patient’s Methods and Materials

### 2.1. Sources of Data

We obtained the gene expression profiles and related clinical information (patient ID, patient age, patient gender, grade, pathological stage, OS time, and survival status) of kidney renal clear cell carcinoma (KIRC) cohorts from The Genomic Data Common (TCGA) (https://portal.gdc.cancer.gov/, accessed on 5 August 2022). We included coding genes (19,658 genes) from fragments per kilobase of exon per million mapped (FPKM). In the study of gene expression, these were matched with the ICI genes that were obtained from new ICI databases (the checkpoint therapeutic target database (CKTTD) and recently published new ICI genes list) [24,25]. For validation of our gene signature, we used the gene expression profile of 604 ccRCC patients from the GEO database (GSE167093) [26]. We do not need to obtain approval consent when accessing deidentified public data.

### 2.2. Prognostic Genes Score Construction

After the matching of ICI genes and the coding genes of the KIRC cohort, in order to choose the survival-related genes, we used univariate analysis using the R package “survival”, and *p* < 0.05 as screening criteria. The hazard ratio (HR) and 95% confidence interval (CIs) were tested by the Wilcoxon rank test. After that, we applied the multivariable cox hazard model and Wilcoxon rank test for the calculation of HR with 95% CIs and the results were used to construct a prognostic index (PI) to evaluate the prognostic risk of ccRCC by using the R package “survival”, and “survminer”. Each patient’s risk score (RS) was calculated according to the PI by following the formula RS = Σexpgenei* βi. KIRC patients were then divided into high and low risk groups based on the median value of risk score. By applying the Kaplan–Meier (KM) method and long-rank test, the survival differences between the two groups were determined. To unveil the prognostic ability of risk score signature, univariate and multivariable cox hazard models were done with clinical variables. The receiver operating characteristic (ROC) curve was plotted by the R packages “survival”,” ROCR”, and “timeROC” and it was used to estimate the area under the curve (AUC). In addition, a nomogram for the prediction ability of our prognostic genes was constructed that resulted from a multivariable cox hazard model using the R packages “survival” and “regplot” to predict patient survival at 1, 3, and 5 years. The correlation of the resulting risk score genes from the multivariable cox hazard model with clinical characteristics, including age, was evaluated using The Kruskal–Wallis test. The Spearman correlation test was also performed to analyze the correlation between gene expression and age. Finally, the survival of the resulting genes was investigated according to their median expression using a human protein atlas (https://www.proteinatlas.org/, accessed on 5 August 2022).

### 2.3. Enrichment Pathways, and Protein–Protein Interaction

To understand the underlying mechanism, we utilized the R package “org.Hs.eg.db” to convert the IDs of ICI matched genes, and “clusterProfiler”, “enrichplot”, and “ggplot2” [27] for performing enrichment analysis of Gene Ontology (GO) and Kyoto Encyclopedia of Genes and Genomes (KEGG) pathways. We used the program Cytoscape (https://cytoscape.org/, accessed on 5 August 2022) to build as well as illustrate the PPI of ICI genes of the CKTTD database and recent ICI gene studies. Finally, the PPI network was done between risk signature genes and the list of aging genes that were recruited from GSEA (http://www.gsea-msigdb.org/gsea/msigdb/search.jsp, accessed on 5 August 2022).

### 2.4. The Immune Functional Types and Immune Cell Infiltrations

Immune cells may be found in all of the different risk categories. “limma”, “ggpubr”, and “reshape2” packages using the R software were used to discover the relationship of risk score with respect to sixteen immune cells and thirteen immune-related function types. In addition, it enabled the visualization of the association between their expression and the amount of immune infiltration in KIRC.

### 2.5. Statistical Analysis

By conducting a Chi square test, the clinical characteristics of the TCGA cohort between young and old groups were analyzed. We identified prognostic genes using the univariate Cox hazard analysis method. After that, by using the survival package in the R language program, we accomplished a multivariable Cox hazard model that unveiled a score signature. In addition, correlation, one-way ANOVA on rank, and independent tests were done. All statistical methods are carried out using R version 4.0.4 and SPSS version 26. We consider *p* value < 0.05 and a limit of 0.0001 as statistically significant.

## 3. Results

### 3.1. Study Cohort Clinical Characteristics

A total of 528 KIRC patients with known survival were retrieved from the TCGA database (Table 1). In the study of clinical related variables for the TCGA ccRCC cohort and comparing between the young and old groups. After we evaluated them, we found there were no significant between stage and grade (*p* = 0.26, and *p* = 0.62), respectively. Theoretically, the grade and stage are the main variables that could affect the survival of cancer patients. Thus, this supports our hypothesis of the difference that might be seen in the study of genomic level.

### 3.2. Prognostic Gene Signature Construction

The ICI gene list was matched with the transcriptomic gene expression of KIRC cohort patients and then we did a univariate analysis that resulted in 28 genes with high significant difference in high vs. low expression *p* < 0.05; HAVCR2, PDCD1, LGALS9, TNFRSF12A, HLA-E, HLA-G, BTNL9, TDO2, CD276, TIGIT, TNFRSF8, TNFRSF9, CD44, TNFSF15, KIR2DL4, PVR, TNFSF4, TNFSF14, CEACAM1, TNFRSF25, NECTIN2, LAG3, CTLA4, LAIR1, NRP1, CD80, HHLA2, and TNFRSF18 (Appendix A). In addition, multivariable Cox hazard model analysis uncovered 14 ICI genes that were considered risk gene signatures (HHLA2, TNFRSF12A, HLA-G, NECTIN2, TNFRSF25, TNFSF14, LAIR1, TNFSF15, TNFSF4, KIR2DL4, PDCD1, LAG3, LGALS9, and PVR). Finally, the top nine significant genes (*p* < 0.05) were chosen to be risk score genes (HHLA2, TNFRSF12A, HLA-G, NECTIN2, TNFRSF25, TNFSF14, LAIR1, TNFSF15, and TNFSF4) (Appendix A).

The capacity for each ICI gene signature-related overall survival (OS) was assessed using the KM survival analysis method to predict the prognosis of KIRC patients. Those in high risk score groups have significantly lower survival status than lower risk score patients with *p* < 0.001 (Appendix A).

This study then examined the nine genes signature prediction value using a nomogram and ROC curve (Appendix A). As a result, the risk scores of KIRC cases are classified into two categories: the mortality rate in the higher risk score group was considerably higher than the lower group (Figure 1). We have identified that the majority of risk score genes (TNFSF14, LAIR1, NECTIN2, TNFRSF12A, and TNFSF4) are oncogenes while (HHLA2, and TNFSF15) are oncosuppressor genes *p* < 0.05 (Figure 2).

### 3.3. Genes Involved in Ageing Disparity

After doing an analysis using the multivariable Cox hazard model for clinical characteristics that are connected with a risk score, it was determined that there is an accurate approach for predicting prognosis. In addition to the risk score, which was revealed to be an independent variable (HR = 1.241. *p* = 0.0001, CI = 1.81–1.304), the researchers discovered that age was also a statistically significant independent factor (HR = 1.518. *p* = 0.009, IC = 1.110–2.076) (Figure 3). The age group clustering with our risk score genes and clinicopathological characteristics are shown in Appendix A.

TNFSF15 gene expression was associated with aging in KIRC patients, as shown by an independent test in which the gene expression level was higher in younger than older patients (*p* = 0.0001) (Figure 4A). Furthermore, the correlation analysis of TNFSF15 expression in the KIRC cohort with aging also demonstrated a negative correlation (*p* < 0.001, r = −0.17) with a linear regression equation, Y = −0.005785 × X + 1.077. An external validation of the TNFSF15 was obtained from GEO (GSE167093) using a sample of 604 patients that showed significant expression differences between young and elderly patients (*p* = 0.007). We further analyzed the TNFSF15 expression among age groups by using the UALCAN database (http://ualcan.path.uab.edu/, accessed on 5 August 2022) which demonstrated decreased median expression of the TNFSF15 gene when patients were getting older (Figure 4).

Then, we evaluated the predictive model of the genes risk score, patient age, and stage by an innovative nomogram, which has been shown to identify mortality risk with great sensitivity across 1, 3, and 5 year AUC (85%, 81%, 81%), respectively (Figure 5). Further analysis of the risk score genes with clinical variables using independent tests TNFSF15 also showed significant differences between the stage groups (Figure 6).

### 3.4. GO, KEGG, and PPI Network

In terms of GO analysis, the regulation of cellular senescence, positive regulation of cellular senescence, peptide antigen binding, immune receptor activity, and tumor necrosis factor receptor superfamily binding were key biological processes involving genes that were related with matched ICIs in the KIRC cohort. KEGG analysis revealed, in addition to the cellular senescence pathway, the pathways of interactions between cytokines and their receptors, Th1 and Th2 cell differentiation, as well as those involving PD-L1 expression and the PD-1 checkpoint pathway in cancer were the most enriched pathways (Appendix A). In the context of TNFSF15 gene expression, both ICIs and aging genes exhibited PPI interaction (Appendix A).

### 3.5. Identification of Different Risk Groups Associated with Immunity

Immune cell infiltration showed that the infiltration scores of immune cells of interstitial dendritic cells (iDCs) and mast cells were significantly higher in the low risk group than high risk group *p* < 0.05; while macrophages, plasmacytoid dendritic cells (pDCs), T-helper cells, T follicular helper cells (Tfh), Th1-cells, Th2-cells, and tumor infiltrating lymphocyte (TiL) cells were significant higher in the high risk group than the lower risk group (Appendix A). In immune functional analysis, there were significant infiltration scores of type-II-IFN-response in the lower risk group than the high risk group. On the other hand, the antigen presenting cell (APC), APC-co-inhibition, APC-co-stimulation, CCR chemokine, checkpoint, inflammation-promoting, parainflammation, and T-cell-co-stimulation showed higher infiltration scores in the high risk group than in the low risk group all *p* < 0.05 (Appendix A).

## 4. Discussion

The proportion of the population comprised of people over 65 who have survived cancer has increased, and the life expectancy gap between those with and without cancer has widened in the past few years [28]. Patients over 60 years account for almost 50% of all newly diagnosed cancers. In addition, those over the age of 65 make up the largest demographic category of tumor-related fatalities [20,29]. Due to the fact that the immune system also declines with age, which may be associated with the aggressiveness of cancers, immunosenescence has lately emerged as a hot topic of medicinal debate. In addition, the approval of innovative immunomodulatory antibody treatments that depend on the patient’s immune system to treat cancer generated evolving new research and discussion [30]. This research attempts to find novel biomarkers that may be accountable for the disparities in age and survival prognosis of ccRCC in the era of ICIs.

The checkpoint therapeutic target database (CKTTD) is a repository of checkpoint protein targets for which experimental evidence has been validated and curated using an advanced text-mining technique from 10,649 papers. Moreover, it is a well-kept resource that is applicable to available checkpoint targets and cancer immunotherapy treatment. CKTTD is the first online database of its sort, and any researcher working on cancer immunotherapy and checkpoint molecule drug development gets free access to this information. In addition, it gives an evidence-based summary of checkpoint targets and potential therapy options for checkpoint molecules or their regulatory pathways. By utilizing CKTTD and applying available bioinformatics techniques, checkpoint-targeted therapy can be developed. Combining targeted medicine that is aimed at the microenvironment for cancer with conventional treatment such as chemotherapy and radiotherapy might enhance tumor patient’s efficacy in the practical field. Researchers anticipate the development of additional therapies to tackle checkpoint components would enhance the clinical results for patients and eventually lead to cancer eradication [24,25,31]. The majority of findings for elderly and ccRCC patients, whether in localized or distant areas during the ICI and non-ICI era, are still in their early stages [32,33,34]. In the clinical context of RCC aging, several reports revealed that the prognosis for individuals over 65 years old with RCC is significantly worse than for those under 65 years [35,36,37]. According to the finding of this cohort study, the prognosis for older patients is markedly worse than that for younger patients. We reported a novel, significant risk score for genes that applied to the most recent ICI genes database. This score will aid in risk identification and prediction for ccRCC patients. In this cohort as a whole, the effects of aging were readily apparent, and they were notably more severe in the elderly than in the younger patients. Additionally, we examined the risk genes, correlated them with age, and identified the oncosuppressor genes that are less prevalent in older ccRCC patients than in younger patients. By studying of the gene-related pathways and ontology, they were linked with immune related pathways and cell aging.

Along with TNFSF14, LAIR1, NECTIN2, TNFRSF12A, HHLA2, and TNFSF4, TNFSF15 was found to be significantly linked and adversely associated with age in ccRCC patients in TCGA and GEO datasets. Our risk score signature could serve as an important biomarker in the coming years as we noticed the expression in old people for ICI genes was higher than in young people, particularly in the immunotherapy era. The literature has reported this new gene TNFSF15 and its role in angiogenesis and inflammation but does not describe it is clinical efficacy in RCC and/or associated immune cells, principally in aging. Multifaceted cytokine tumor necrosis factor superfamily-15 (also known as VEGI15 or TL1A16) is mostly released by endothelial cells in existing blood channels [38], where it then leads to the inhibition of new blood vessels [39,40,41]. Some studies have shown that high TNFSF15 levels in cancers decrease the growth of the tumor [42,43]. In addition, TNFSF15 leads to the activation of both Th1/Th2 immune responses in various inflammatory processes to help maintain inflammation and immune homeostasis [44,45]. Moreover, TNFSF15 has a key function in the regulation of immune cell activities, which include increasing T-cell activation, maturation of dendritic cell, and increasing the NK cell toxicity as well as enhance macrophage capacity to defeat microbes [46,47,48,49]. It is interesting to note that TNFSF15 gene expression is almost totally down-regulated in proliferative endothelium and in tumor vasculatures and that may increase the maturation and polarization of macrophages toward a tumor-killing M1 type [44]. TNFSF15 is one of the TNFα genes that induced cell senescence via the JAK/STAT pathways [50]. The findings of our study identify certain genes and gene pathways that might be responsible for the decrease in the average life span that is associated with aging.

The expression of TNFSF15 in a variety of immune cells as well as survival is investigated in this study. As a new prognostic model, the TNFSF15 prognosis level and related age were considerably demonstrated. Furthermore, we confirmed TNFSF15 expression in the GEO database and studied the expression value, which demonstrated a substantial rise of the expression in younger individuals (<65 years) when compared to older people, and the distinction was obvious. In addition, we investigated the factors that may be responsible for these shifts and think about the fascinating research questions that may be asked in the future about elderly persons.

Our research, similar to other retrospective cohort studies, has several limitations. It is difficult to exclude confounding factors due to the retrospective character of this research, and they may be connected to the prognosis of affected patients. The deep learning and biological algorithm approaches that were obtained from the GEO and TCGA datasets have recently needed to be confirmed using in-depth investigation. In addition, the ccRCC is engaged in too many complex processes and mechanisms, but we rely on mRNA gene expression and protein expression in our work. Larger clinical multicenter randomized studies have to be conducted to determine how the genes in ccRCC may contribute to the prognosis and treatment of older patients in the ICI era.

## 5. Conclusions

The study’s goal was to assess the predictive value of a new ICI gene-related risk score that was created using a validated ICI gene list in patients with ccRCC. Moreover, we identified a new ICI gene (TNFSF15), an oncosuppressor gene that is associated with age-related survival inequalities that are attributed to immunological state and its senescence, which may be associated with the development of aggressive behavior and carcinogenesis in older patients. From this, we can understand how the gene’s products function and how ccRCC patients adapt to aging in order to identify potential therapeutic targets will need extensive investigation and large cohort sets.

## Figures and Tables

**Figure 1 cells-11-03641-f001:**
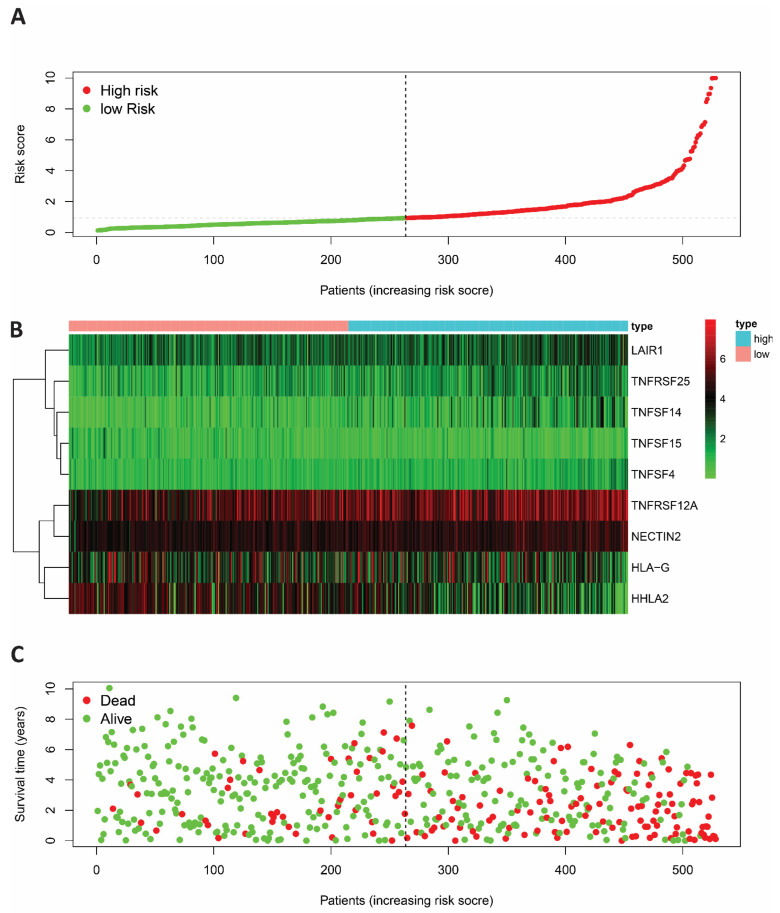
(**A**) The risk score distribution of the genetic signature of ccRCC patients in high vs. low risk groups (green line = low risk; red line = high risk). (**B**) The heatmap of signature genes in the high and low risk groups. (**C**) The risk score signature and survival status of ccRCC patients with a higher number of deaths in the high risk group than in the lower risk group (green dots = alive, red dots = dead).

**Figure 2 cells-11-03641-f002:**
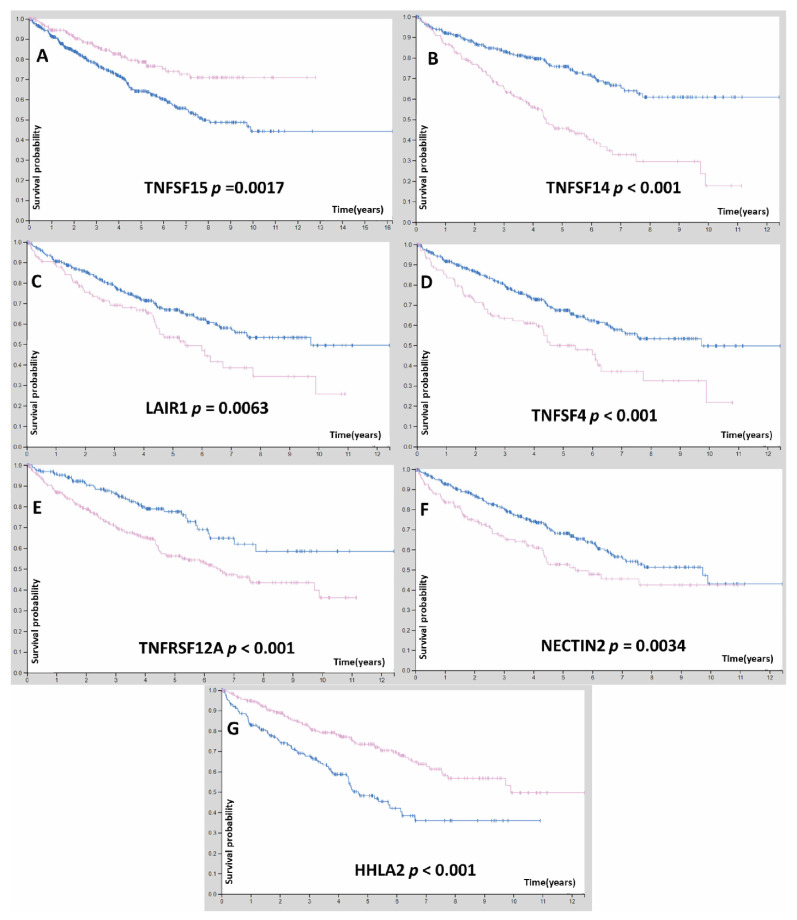
(**A**–**G**) Analysis of median survival differences of ccRCC patients of risk score signature genes (*p* < 0.05). TNFSF15 represented as oncosuppressor where high expression linked with higher survival than lower expression. blue line means low expression while red line represents high expression.

**Figure 3 cells-11-03641-f003:**
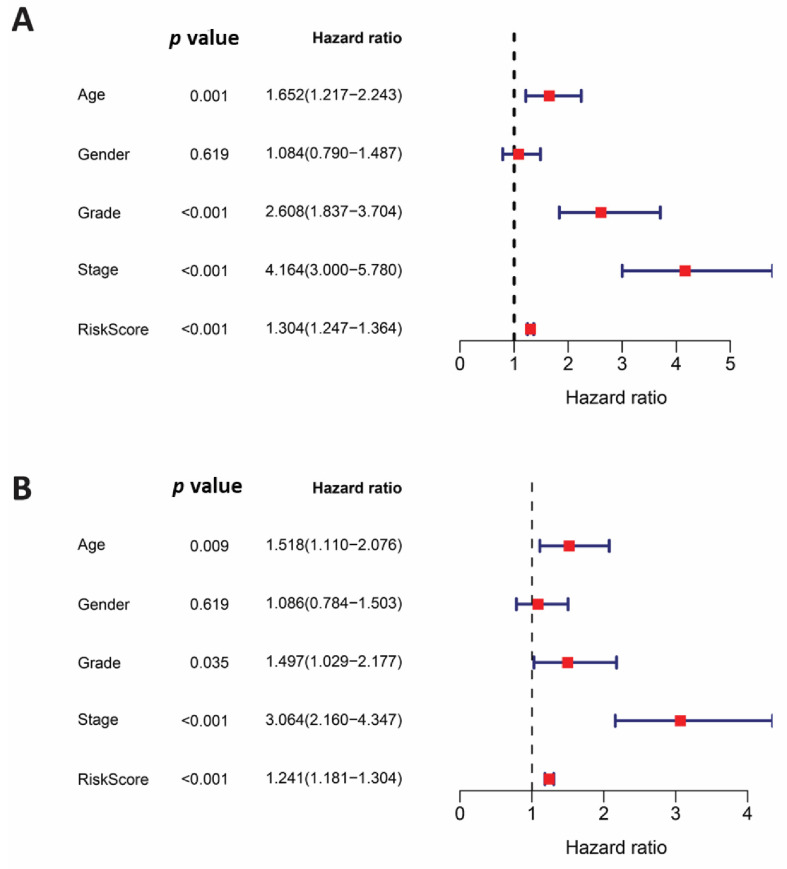
Adjusted risk score signature with clinical variables by using univariate and multivariable Cox hazard model analysis. (**A**) Analysis of univariate data shows that risk score, age, stage, and grade were significant. (**B**) By using multivariable Cox hazard model analysis, age is still an independent prognostic indicator in addition to the risk score *p* = 0.009 (HR = 1.5 (CIs = 1.1–2)) and *p* < 0.001 (HR = 1.24 (Cis = 1.18–1.3)), respectively.

**Figure 4 cells-11-03641-f004:**
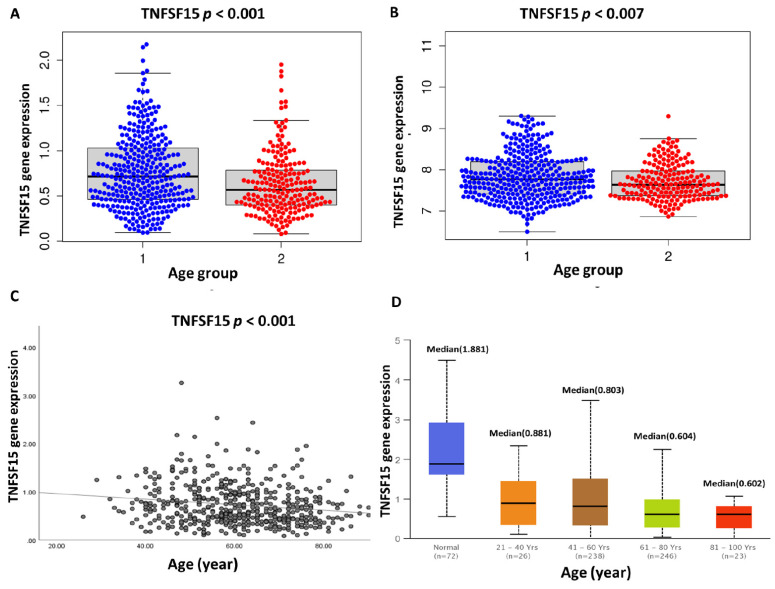
TNFSF15 gene expression between the young and old age groups. (**A**). TCGA ccRCC cohorts show significantly higher expression of the TNFSF15 gene in young groups than in older groups *p* < 0.001. (**B**). TNFSF15 gene expression in the GEO ccRCC patient cohort showed high expression in the young compared to the old group *p* = 0.007. (**C**). TCGA ccRCC TNFSF15 gene expression showed a negative correlation with age *p* = 0.0001. (**D**). The TNFSF15 gene expression in normal ccRCC vs. ccRCC-related cancer tissues; median TNFSF15 gene expression decreased with increasing age in the cancer patient. (1) = young group <65 years and (2) = old group ≥65 years.

**Figure 5 cells-11-03641-f005:**
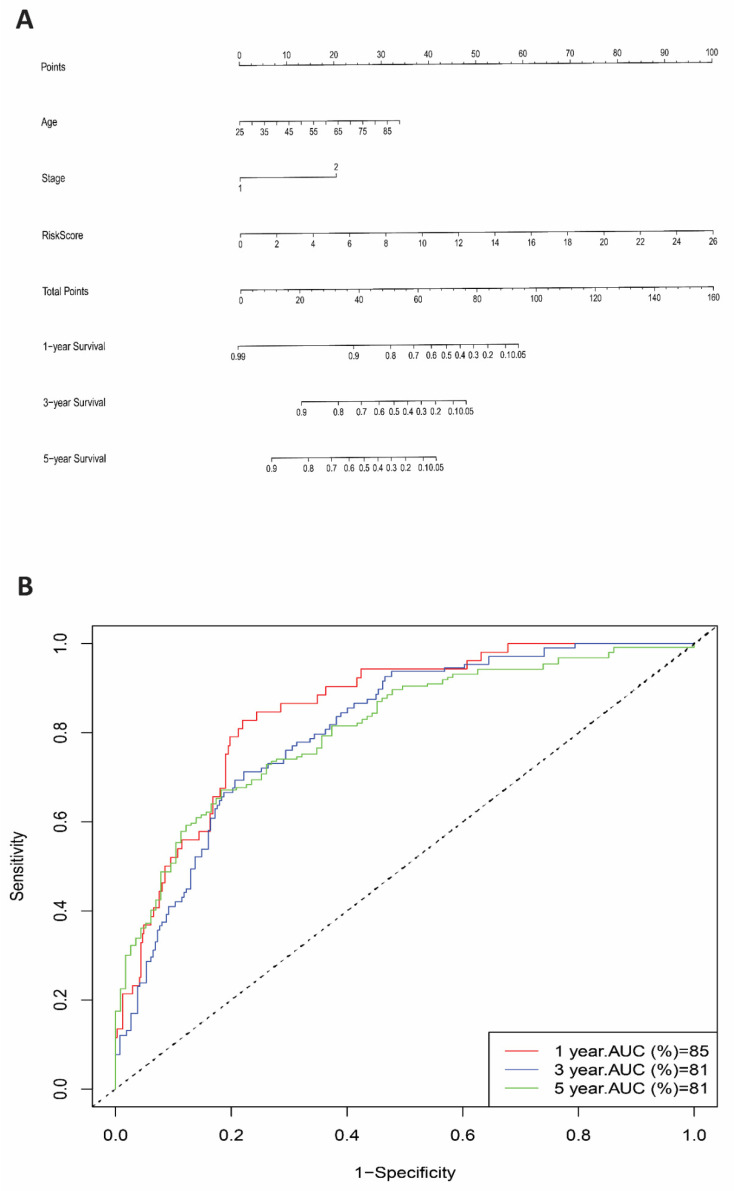
Risk score signature and clinical variable prediction values. (**A**) Nomogram prediction of 1-yr, 3-yr, and 5-yr survival based on risk score, age, and stage. (**B**) ROC curve that showed high predictive accuracy of risk score in 1-yr, 3-yr, and 5-yr (AUC = 85%, 81%, and 81%, respectively). (stage: (1) = stage I, II, (2) = stage III, IV).

**Figure 6 cells-11-03641-f006:**
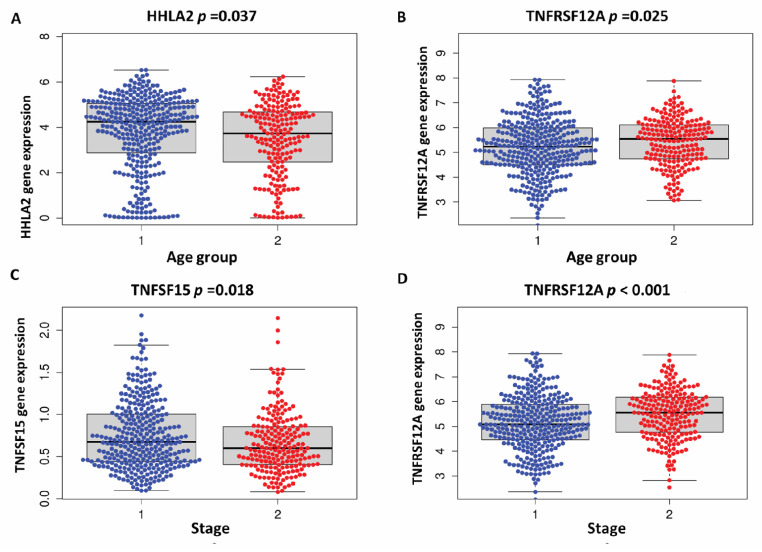
The median gene expression of risk score signature genes with clinical data. (**A**) HHLA2 gene expression show significantly high expression in the young compared to the old age group *p* = 0.037. (**B**) TNFRSF12 gene expression showed significantly high expression in the old compared to the young age group *p* = 0.025. (**C**) TNFSF15 gene expression showed significantly high expression in the stage I, II compared to stage III, IV *p* = 0.018. (**D**) TNFRSF12 gene expression showed significantly high expression in the stage III, IV compared to stage I, II *p* < 0.001. (age (1) = young group <65 years and (2) = old group ≥65 years, and (stage: (1) = stage I, II, (2) = stage III, IV).

**Table 1 cells-11-03641-t001:** Clinicopathologic features of the KIRC cohort between young versus old age groups.

Clinicopathologic Features	Young Age Group <65 Years; Number (%)	Old Age Group ≥65 Years; Number (%)	*p*-Value
All cohort (528)	331 (62.6)	197 (37.3)	
Gender			<0.001
Male	234 (70.6)	109 (55.4)	
Female	97 (29.4)	88 (44.6)	
Stage			0.26
(I, II)	210 (63.4)	111 (56.4)	
(III, IV)	119 (36)	85 (43.1)	
Missed data	2 (0.6)	1 (0.5)	
Grade			0.62
G I, GII	155 (46.8)	86 (43.6)	
G III, G IV	172 (51.9)	107 (54.2)	
Missed data	4 (0.3)	4 (0.2)	
Risk			0.03
Low	182 (55)	82 (58.4)	
High	149 (45)	115 (41.6)	
Survival status			<0.001
Dead	86 (26)	79 (40.2)	
Alive	245 (74)	118 (59.8)	

## Data Availability

All data in this study are open for all without need for patient consent and no require access approval. It is available in the GDC portal repository (https://portal.gdc.cancer.gov/, accessed on 5 August 2022).

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
