# Peer review of "Immune Checkpoint Inhibitor (ICI) Genes and Aging in Clear Cell Renal Cell Carcinoma (ccRCC): Clinical and Genomic Study"

_cells, 2022, doi:10.3390/cells11223641_

Round 1

Reviewer 1 Report

Dr. Al-Danakh et al investigate the role of ICI genes in clear cell RCC patients. There is some potentially valuable data, but it is extremely difficult to understand. Many conclusions unsupported by data or tables. My concerns are detailed below.

Major points:

1.       Patients state they conducted a “clinical and genomic study”. What “clinical” study did they conduct? If only a genomic study was performed, the title should be revised.

2.       Materials and Methods is extremely short and insufficient. There are only several lines for each portion and are difficult to understand. Please provide sufficient details so that results can be reproduced.

3.       Baseline data is not provided for much of the analyses, making it impossible for the reader to reproduce results. Please provide summaries of baseline data of the 528 ccRCC patients.

4.       Please provide analysis for elderly vs. young patients. What is the cutoff age? I assumed it is 65, but this should be stated under methods. How many patients were in each group? What were the baseline characteristics for each? Without this data, the reader cannot evaluate the presence of confounders.

5.       Were ICIs used in all 528 patients? If not, this study does not evaluate the relationship between ICIs and aging, as suggested in the title. Then, the study also has nothing to do with ICIs, and only is related to “ICI genes.” The title is again misleading.

6.       What was the statistical methods used for figures 4 and 6? This is not described under methods.

7.       TNFSF15 gene expression was significantly lower in elderly ccRCC patients than in younger patients. However, the scatter plot in Figure 4B looks very scattered, with no clear trend for lower expression in older patients. If there really is a strong correlation, please provide the regression line and equation.

Minor points:

1.       Language errors throughout.

2.       Captions in Figure 2 are too small to read.

3.       Sup. Figure 3 is too small to read.

4.       Suppl Table 1: is this 19-page table really necessary? A summary should be provided instead of raw data.

5.       Suppl Table 1: What are futime and fustat? These are not explained anywhere. Please explain all abbreviations.

Author Response

Comments and Suggestions for Authors

Dr. Al-Danakh et al investigate the role of ICI genes in clear cell RCC patients. There is some potentially valuable data, but it is extremely difficult to understand. Many conclusions unsupported by data or tables. My concerns are detailed below.

Major points:

Comment 1

Patients state they conducted a “clinical and genomic study”. What “clinical” study did they conduct? If only a genomic study was performed, the title should be revised.

Reply to comment 1.

Thank you for your comment, the clinical characteristics of KIRC have been summarized in a table 1 in addition to aging clustering heatmap between young and elderly groups in supplementary figure 3 you may see table 1 and supplementary figure 3  

Comment 2

Materials and Methods is extremely short and insufficient. There are only several lines for each portion and are difficult to understand. Please provide sufficient details so that results can be reproduced.

Reply to comment 2.

It’s fixed now as you recommend; material, methods and result sections have been explained well and expanded in details and understandable manner you may see lines (95-139, and 149-212).  May you have further consideration it is our pleasure.

Comment 3

Baseline data is not provided for much of the analyses, making it impossible for the reader to reproduce results. Please provide summaries of baseline data of the 528 ccRCC patients.

Reply to comment 3.

 We are sorry for giving raw data in previous submission, like your advice the clinical characteristics of KIRC have been summarized in a table 1 and supplementary figure 3 you may referred to table 1 and supplementary figure 3.

comment 4

Please provide analysis for elderly vs. young patients. What is the cutoff age? I assumed it is 65, but this should be stated under methods. How many patients were in each group? What were the baseline characteristics for each? Without this data, the reader cannot evaluate the presence of confounders.

Reply to comment 4.

 Thank you very much for such valuable comment, we have take it in consideration and add it in table 1 and heatmap clustering in supplementary figure 3. you may see table 1 and table 1 and supplementary figure 3.

comment 5

Were ICIs used in all 528 patients? If not, this study does not evaluate the relationship between ICIs and aging, as suggested in the title. Then, the study also has nothing to do with ICIs, and only is related to “ICI genes.” The title is again misleading.

Reply to comment 5.

 Yes, you are right, this was miswritten and term genes added to the title as you recommend you may see title line 2,3.

comment 6

.       What was the statistical methods used for figures 4 and 6? This is not described under methods.

Reply to comment 6.

 It was done by using the Kruskal-Wallis test which mentioned in the method part you may see line (122-124)

comment 7. 

      TNFSF15 gene expression was significantly lower in elderly ccRCC patients than in younger patients. However, the scatter plot in Figure 4B looks very scattered, with no clear trend for lower expression in older patients. If there really is a strong correlation, please provide the regression line and equation.

Reply to comment 7

 Actually, it is highly significantly seen with P<0.0001 with Equation   Y = -0.005785*X + 1.077 that had been rephrased in the result section you may see line (181-183)……in addition we have been added another figure that show median expression differences between each age subgroups you may see figure. 

Minor points:

 comment 1.

              Language errors throughout.

Reply to comment 1.

 In this stage we have revised it comprehensively by helping of native English speaker. you may see revised manuscript, you may have further consideration  it is our pleasure.

comment 2

Captions in Figure 2 are too small to read.

Reply to comment 2

 It is fixed in this stage with clear unified caption you may see figure 2

comment 3

Sup. Figure 3 is too small to read.

Reply to comment 3

Thank you for this note, we have been replacing GO and KEGG pathways to supplementary table 1 that is clear and we think is more informative, you may have consideration it is our pleasure. you may see lines and supplementary table 1.

       comment 4

Suppl Table 1: is this 19-page table really necessary? A summary should be provided instead of raw data.Suppl Table 1: What are futime and fustat? These are not explained anywhere. Please explain all abbreviations.

Reply to comment 4

Thank you again, we summarized it in understandable way as you recommend and replace futime and fustat with survival time and survival status respectively you may see table 1

may you have further consideration,it is our pleasure.

Warm regards

Reviewer 2 Report

In their study, Al-Danakh et al. assess the risk score of age-related immune checkpoint inhibitors (ICIs) gene expression in ccRCC patients. The ICI gene TNFSF15 was specifically defined as a tumor suppressor gene and was connected to aging. According to the authors, this score will contribute to risk identification and prediction for these patients. Basically, the manuscript was well organized.

I suggest the authors extend in the text the description relating to the results in the paragraphs:

-         “ Go, KEGG, and PPI network” (page 9, line 195-202)

-        “The immune functional types and immune cell infiltrations” (page 9, line 203-211). 

Author Response

Comments and Suggestions for Authors

In their study, Al-Danakh et al. assess the risk score of age-related immune checkpoint inhibitors (ICIs) gene expression in ccRCC patients. The ICI gene TNFSF15 was specifically defined as a tumor suppressor gene and was connected to aging. According to the authors, this score will contribute to risk identification and prediction for these patients. Basically, the manuscript was well organized.

Comment 1

I suggest the authors extend in the text the description relating to the results in the paragraphs:

-         “ Go, KEGG, and PPI network” (page 9, line 195-202)

Reply to comment 1

Thank you for your valuable comment.in addition to GO, and PPI network explanation, we have expanded description of the enrichment pathways as you recommend; you may see lines (193-201)

Comment 2

-        “The immune functional types and immune cell infiltrations” (page 9, line 203-211). 

Reply to comment 2

We now explained in detail the immune functional types and cell infiltration like your advice. You may see lines  (203-212)

may you have further consideration, it is our pleasure.

Best regards

Reviewer 3 Report

The study is interesting and original. The method is simple and easy to understand. The bibliography is recent and complete. In the text and as well as in the figures are very few correction to be made:

line 58: a eternal please replace with an eternal

line 66, please remove in function written twice

line 100, please, extend the description of KIRC

Please insert the description of the Figure 2 in the text, Has it become a supplementary Figure?

Figure 2 please increase the size of the characters, both the names of the proteins that p

Figure 4A and 4B, please correct the values of p and increase the size of A, B, C

Figure 6, to make the data more clear, please move stage above and age below

A curiosity: have you already applied the same method of studies for other tumor histotypes?

Author Response

Comments and Suggestions for Authors

The study is interesting and original. The method is simple and easy to understand. The bibliography is recent and complete. In the text and as well as in the figures are very few corrections to be made:

Comments 1.

line 58: a eternal please replace with an eternal

Reply to Comment1.

Thanks for your notice , it is fixed you may see line(70)

Comments 2.

line 66, please remove in function written twice

Reply to comment 2.

It has been removed, you may see line (77)…

Comment 3.

line 100, please, extend the description of KIRC

Reply to comment 3.

In addition to the writing of KIRC abbreviation, we have checked all abbreviation and fixed in first appearance in the whole manuscript, you may see line (97)

Comment 4.

Please insert the description of the Figure 2 in the text, Has it become a supplementary Figure?

Reply to comment 4.

 Thank you for your valuable note, it was missed unintentionally, now it is fixed you may see line(170)

Comment 5.

 Figure 2 please increase the size of the characters, both the names of the proteins that p

Reply to Comment 5

 Yes, you are right, it is fixed and updated to clear unified version, you may see figure 2

Comment 6.

Figure 4A and 4B, please correct the values of p and increase the size of A, B, C

Reply to Comment

Ok, they were fixed and replaced with clear version

Comment 7

 Figure 6, to make the data clearer, please move stage above and age below

Reply to Comment 7.

 As you recommend, it has been moved into right place, you may see figure 6

 Comment 8.

A curiosity: have you already applied the same method of studies for other tumor histotypes?

Replay to Comment 8.

Actually we investigate KIRC that represent majority of cases in TCGA about RCC, other histotypes have less number samples and we thought that will invaluable results .

May you have further consideration, it is our pleasure 

Warm regards

Round 2

Reviewer 1 Report

I believe the authors have significantly improved their manuscript. 

Just a few comments.

1. Thank you for adding Table 1. Could you add p-values to this table? This will show that differences in ICI genes were not due to differences in baseline factors other than age, such as cancer stage. I would also add a sentence in the manuscript to explain that there were no significant differences in baseline characteristics between the young and old groups.

Also, do you have information on how many patients in each group received ICIs for treatment? It would be interesting to see if that had any impact on survival in this population or had any relationships to ICI genes found. (This would be a good addition to the manuscript but is not essential.)

2. Figure 4D: what is "normal" in the first column? This needs to be clarified. Also, the x-axis legend "Age (year)" covers part of the graph text.

3. ULCAN-->UALCAN

4. Figure 5 legend says: "(A) Nomogram prediction of risk score, age, and stage in 1,3 and 5 years".  This is not accurate. I believe it should be "Nomogram prediction of 1-yr, 3-yr, and 5-yr survival based on risk score, age, and stage."

Also, why does the nomogram include both "risk" and "risk score"? Do both need to be included? Because "risk" is based on "risk score", it looks like the nomogram double counts the risk score. 

5. The labels in Figure 2 are too small to read.

6. There has been an improvement, but there are still some English errors that need to be corrected, especially in figure legends.

Author Response

Dear Professors;

Thank you dear editor and dear reviewer 1 for giving us this opportunity to reply for reviewer1 comment about our manuscript   entitled ( Immune checkpoint inhibitors (ICIs) genes and aging in clear cell renal cell carcinoma (ccRCC): clinical and genomic study) , manuscript ID: cells-1987578.

Reviewer1

Comment 1

Thank you for adding Table 1. Could you add P-values to this table? This will show that differences in ICI genes were not due to differences in baseline factors other than age, such as cancer stage. I would also add a sentence in the manuscript to explain that there were no significant differences in baseline characteristics between the young and old groups.

Reply to comment 1

Thank you for comment, while there was no significant seen in the important clinical characteristics (stage, grade). We have followed your recommendation and did chi square test, and we added to method and result section. You may see lines (143,144; and lines 152-158)

Comment 2

Also, do you have information on how many patients in each group received ICIs for treatment? It would be interesting to see if that had any impact on survival in this population or had any relationships to ICI genes found. (This would be a good addition to the manuscript but is not essential.)

Reply to comment 2

Actually, The TCGA data does not have this information.

Comment 3

Figure 4D: what is "normal" in the first column? This needs to be clarified. Also, the x-axis legend "Age (year)" covers part of the graph text.

Reply to comment 3

Fixed as you recommend may you see figure 4D legend.

Comment 4

ULCAN-->UALCAN

Reply to comment 4

Done thanks for your notice dear professor.(line 196)

Comment 5

 Figure 5 legend says: "(A) Nomogram prediction of risk score, age, and stage in 1,3 and 5 years".  This is not accurate. I believe it should be "Nomogram prediction of 1-yr, 3-yr, and 5-yr survival based on risk score, age, and stage."

Reply to comment 5

It is fixed as you recommend, you may see figure 5 legend.

Comment 6

 why does the nomogram include both "risk" and "risk score"? Do both need to be included? Because "risk" is based on "risk score", it looks like the nomogram double counts the risk score. 

Reply to comment 6

We appreciate you rising this important point; it has been fixed as you advice. May you see  Figure 5 A

Comment 7

The labels in Figure 2 are too small to read.

Reply to comment 7

Fixed and all figure labels unified and become more clear; you may see figure 2.

Comment 8

There has been an improvement, but there are still some English errors that need to be corrected, especially in figure legends

Reply to comment 8

We have revised it comprehensively , fixed and make further explanation in the figure legends to be more understandable .May you have further considerations it is our pleasure.

Best regards

First author: Abdullah al-danakh

Corresponding author: Deyong Yang